# A Proposal to Classify and Assess Ecological Status in Mediterranean Temporary Rivers: Research Insights to Solve Management Needs

Antoni Munné [1,*], Núria Bonada [2,3], Núria Cid [2,4], Francesc Gallart [2,5], Carolina Solà [1], Mònica Bardina [1], Albert Rovira [1], Clara Sierra [2], Maria Soria [2,3], Pau Fortuño [2,3], Pilar Llorens [2,5], Jérôme Latron [2,5], Teodoro Estrela [6], Arancha Fidalgo [7], Inmaculada Serrano [7], Sara Jiménez [7], Rosa Vega [8] and Narcís Prat [2]

1   Agència Catalana de l'Aigua (ACA), c/Provença, 260, 08036 Barcelona, Spain; csolao@gencat.cat (C.S.); mbardinam@gencat.cat (M.B.); alrovirag@gencat.cat (A.R.)
2   Freshwater Ecology, Hydrology and Management (FEHM) Research Group, Departament de Biologia Evolutiva, Ecologia i Ciències Ambientals, Facultat de Biologia, Universitat de Barcelona, Diagonal 643, 08028 Barcelona, Spain; bonada@ub.edu (N.B.); nuria.cid-puey@inrae.fr (N.C.); francesc.gallart@idaea.csic.es (F.G.); clarasiez@gmail.com (C.S.); mariasoriaextremera@gmail.com (M.S.); pfortuno@ub.edu (P.F.); pilar.llorens@idaea.csic.es (P.L.); jerome.latron@idaea.csic.es (J.L.); nprat@ub.edu (N.P.)
3   Institut de Recerca de la Biodiversitat (IRBio), Universitat de Barcelona (UB), 08028 Barcelona, Spain
4   INRAE, UR RiverLy, Centre de Lyon-Villeurbanne, 5 rue de la Doua CS70077, 69626 Villeurbanne CEDEX, France
5   Surface Hydrology and Erosion Group, IDAEA, CSIC. Jordi Girona 18, 08034 Barcelona, Spain
6   Dirección General del Agua, Ministerio de Transición Ecológica y Reto Demográfico, Plaza San Juan de la Cruz, 28003 Madrid, Spain; testrela@miteco.es
7   Confederación Hidrográfica del Júcar, Av/Blasco Ibañez 48, 46010 Valencia, Spain; Arancha.Fidalgo@chj.es (A.F.); Inmaculada.Serrano@chj.es (I.S.); SaraMaria.Jimenez@chj.es (S.J.)
8   Tecnologías y Servicios Agrarios, S.A., c/del Cronista Carreres, 11, 46003 Valencia, Spain; rvega@tragsa.es
*   Correspondence: anmunne@gencat.cat

**Abstract:** The biomonitoring methods implemented by water authorities are mostly developed for perennial rivers, and do not apply to temporary rivers (TRs). We propose a new classification for TRs to better assess their ecological status. It arises from the LIFE+ TRivers project, which was conducted in the Catalan and the Júcar Mediterranean river basin districts (RBD). The European Water Framework Directive (WFD) provided two systems to set river types (systems A or B from Annex II), which have been officially used by water authorities across Europe to set "national river types" (NRTs). However, essential hydrological variables for TRs are largely omitted. NRTs established according to the WFD were compared with TR categories obtained by using a rainfall-runoff model, "natural flows prescribed regimes" (NFPRs), and with "aquatic phases regimes" (APRs) calculated by using TREHS software. The biological quality indices currently used in Spain, based on macroinvertebrates and diatoms (IBMWP, IMMI-T, and IPS), were compared with a "general degradation" gradient in order to analyze the two TR river classification procedures (NFPR and APR). The results showed that NRTs did not properly classify TRs, and that the APR classification identified ecologically meaningful categories, especially those related to stagnant phases. Four "management temporary river categories" based on APRs are proposed to be used for water managers to properly assess the ecological status of TRs.

**Keywords:** non-perennial rivers; intermittent rivers and ephemeral streams; temporary rivers; river basin management plans; Mediterranean; water framework directive

## 1. Introduction

More than half of the river networks worldwide have temporary (or non-perennial) flow regimes. Temporary rivers (TRs), also known as intermittent rivers and ephemeral streams,

or IRES [1], are fluvial ecosystems that recurrently stop flowing and/or become completely dry at some time [2]. They should not be considered as hydrologically challenged perennial rivers, but as highly dynamic ecosystems with a large variability of flow patterns and shifts between wet and dry phases [3]. TRs predominate in arid climate regions, e.g., [4,5], but they are also present in other climate regions, such as temperate, cold, or polar [6]. However, TRs are underestimated globally in number and relevance [7,8]. This is because TRs are rarely included in flow gauging networks [9], and even when gauging stations are present, they are not designed for measuring low or zero flows [10].

The hydrological complexity of TRs represents a challenge for water managers [11], which has resulted in their exclusion or underrepresentation in most biomonitoring programs, preventing the assessment of their ecological status and the development of potential measures for their restoration and conservation [12–14]. This has been evident in Mediterranean climate regions, where TRs predominate [15–17]. Despite the significant advances in biomonitoring methods for ecological status assessment in Mediterranean climate regions [17–21], they are still not properly considered in river management. The management of TRs is also becoming an issue in other European countries, where these ecosystems are newly documented [22]. A proof of this is the recent SMIRES COST Action (science and management of intermittent rivers and ephemeral streams-CA15113), which developed and compiled methods and tools across Europe to address TRs management [23].

An important step in river management is the classification of rivers into different typologies [24,25]. This is vital for biomonitoring practices, as reference conditions (i.e., those unimpacted or least impacted are used as baseline to assess human impacts) may vary across typologies [26,27]. Hydrologists and ecologists have made several attempts to classify TRs by relating them to the different durations of the dry period or the presence of disconnected pools, resulting in a wide variety of terms and definitions (e.g., near or quasi perennial, intermittent, ephemeral, or episodic) [1]. However, there is still no clear agreement among researchers (ecologists, hydrologists, etc.), and even less among water managers. There are two main reasons for this lack of consensus: the divergence of viewpoints related to this purpose, either for research or management [28], and the lack of enough data and suitable information or guides to properly manage them [23,29]. From a river management perspective, the lack of an adequate TR classification can lead to unreliable assessments and management of these watercourses [11].

In Europe, biomonitoring methods implemented under the water framework directive (WFD) [30] have mostly been developed for perennial rivers. This has resulted in many limitations regarding the management of TRs, especially those related to the reliability of biomonitoring indices in detecting anthropogenic impacts. For instance, biological indices are based on the richness of the aquatic community, and this metric typically declines with increasing flow intermittence, regardless of the anthropogenic impact [17,31,32]. These indices are, therefore, not reliable for assessing the ecological status of TRs, and especially for TRs with ephemeral flow regimes, where aquatic communities can be poor or nonexistent during most parts of the year [33,34]. Other limitations of the WFD implementation in TRs include: (i) the definition of "water body", i.e., the requirement for a river reach to be included within the monitoring network and in river basin management plans (RBMPs); and (ii) the classification of river typologies considering the hydrological variability within TRs, and the subsequent set up of reference conditions. For example, in the case of water body delimitation, TRs are usually omitted in Europe due to their small catchment area and low mean natural flows [35]. Another relevant problem is the lack of flowing water when sampling, resulting in sites that have no water nor biological samples due to a mismatch between the timing of regular samplings conducted by water authorities and the timing of flowing periods in TRs.

The purpose of this paper is to propose a TR classification that enables assessment of their ecological status, for an adequate implementation of the WFD. We incorporated the knowledge from two projects related to the research and management of TRs: the MIRAGE Project (2009–2011) (FP7-ENV-2007-1-Mediterranean intermittent river management) and

the LIFE + TRivers project (2014–2018) (LIFE13 ENV/ES/000341). As a result of the MIRAGE project, a novel approach to characterizing the changing hydrological conditions relevant to aquatic life over time in TRs arose, named "aquatic states" [36]. Later, the LIFE + TRivers project developed a practical method for obtaining TR classifications through different sources of information to: (i) better characterize and classify TRs regimes, taking into account the three main aquatic phases (i.e., a simplification of the above-mentioned "aquatic states"): flowing, disconnected pools, and dry stream beds; (ii) advise about a better temporal schedule for water and biological sampling; and, (iii) assess TRs' hydrological alteration or hydrological quality. This approach was implemented in the open-access TREHS software [37]. Here, we present experience from two Mediterranean river basin districts (RBDs), in which the above-mentioned methods for TR classification were applied, and compared with the traditional WFD classifications. The LIFE + TRivers project was conducted in these two Mediterranean RBDs to test the TR classification methods, and their relationship to biological quality indices. Additionally, a tailored hydromorphological quality (HYMO) index for TRs was applied to analyze the results in ephemeral watercourses. LIFE + TRivers outputs were crucial for establishing a suitable management tool focused on TRs and, therefore, for translating research findings into management practices. Considering that the frequency and extension of flow intermittence is expected to increase in the near future because of climate change [38] and the increasing human water demand [3], the development of methods for managers that enable adequate assessment and biomonitoring is timely.

## 2. Study Area

The TRivers project was conducted in two Mediterranean river basin districts (RBD): the Catalan RBD and the Júcar RBD located in NE and E Spain, respectively (close to the Mediterranean Sea). Both RBDs have a Mediterranean climate, with a precipitation ranging from 400 to 750 mm/year, and highly irregular between years ($\pm$200). Low water flows and dry watercourses are usual, particularly in summer, together with sudden floods after heavy rains.

The Catalan RBD has a total area of 16,438 km$^2$ and includes several small to medium sized basins draining into the Mediterranean Sea, and ranging from 500 to 5000 km$^2$ (Figure 1). The Catalan Water Agency (ACA) is the water authority in charge of managing water supply, urban waste water sanitation, monitoring programs, and drafting the RBMP in the Catalan RBD. The main river basins are those corresponding to the Ter and the Llobregat rivers, which are highly variable seasonally and interannually ($\pm$ 9 m$^3$/s), and have an annual average discharge of 28 m$^3$/s and 19 m$^3$/s, respectively. The remaining basins, such as the Tordera, Besòs, Fluvià, Muga, Francolí, Foix, Gaià, and Riudecanyes, have a total area below 600 km$^2$ each, and a scarce and highly irregular flow, with frequent zero flow reaches in dry periods. The Catalan RBD has a high population density (420 inhabitants/km$^2$), which accounts for the strong urban and industrial pressures and high water withdrawals that exacerbate low water flows [39]. According to the Catalan Water Agency (http://aca.gencat.cat (accessed on 9 March 2021)), a total of 248 river water bodies have been recorded (with an average of 17 km per water body and accounting for 6639 km of river network), of which 136 (37%) were identified as heavily modified by morphological and hydrological human alterations, according to the WFD guidelines.

The Júcar RBD has a total surface area of 44,891 km$^2$ and includes several small to medium sized basins (from 607 to 22,208 km$^2$) that also drain into the Mediterranean Sea. The main river basins are those corresponding to the Júcar, Turia, and Mijares rivers, which have an annual average discharge (in natural regime) of 39, 10, and 8 m$^3$/s, respectively. The Júcar River Basin Authority (CHJ: www.chj.es, accessed on 9 March 2021) is the water authority in charge of drafting the river basin management plan (RBMP). A total of 313 river water bodies were identified in the Júcar RBD (with an average of 16 km per water body), of which 32 (10%) were classified as heavily modified by morphological and hydrological human alterations.

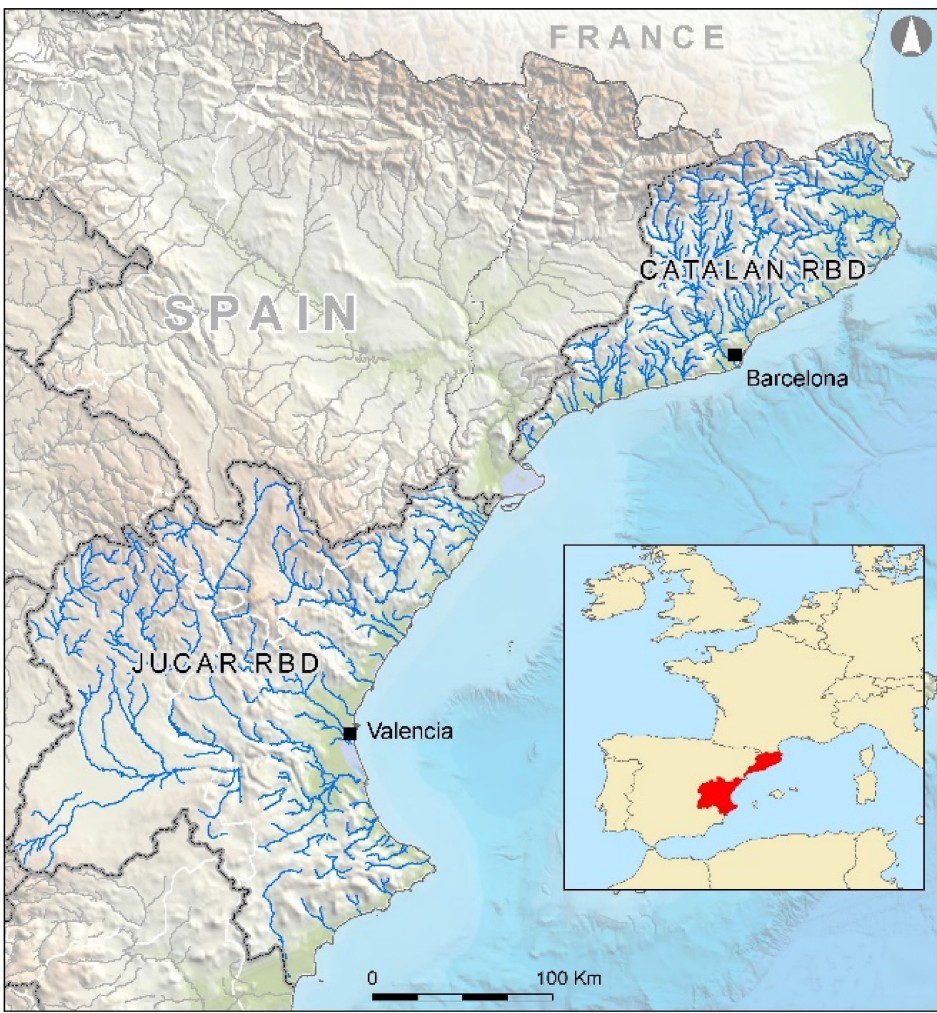

**Figure 1.** Location of the Catalan and Júcar river basin districts (RBDs).

## 3. Temporary River Classification for Management Purposes: Analysis and Results

### 3.1. Comparing Methods for Classifying Non-Perennial Rivers

Defining suitable river types becomes essential classifying the ecological status accurately, and furthermore for applying effective measures to restore and preserve water bodies [40]. Therefore, the EU Water Framework Directive (WFD) (2000/60/EC) proposes defining river types by using either system A or B [30]. Both systems are mainly based on a mix of environmental parameters, which consider water discharge among other variables.

In 2003, a multimetric method was conducted to set river types in the Catalan RBD following WFD system B [41]. A total of 10 river types were defined (Table 1), called "national river types" (NRTs), mainly differing in geology, physical, and climatic characteristics, and water discharge. Shortly after, the same procedure was applied in all Spanish RBDs, defining a total of 33 NRTs [42], of which 14 out of 33 are present in the Júcar RBD. WFD systems A or B have been used "officially" by water authorities to set NRTs so far in the respective European RBMPs in order to assess the ecological status of river water bodies. However, essential hydrological variables, such as temporality, flow variability, and zero flows, were not taken into account, since they were not required in WFD Annex II. Therefore, TRs were largely neglected.

**Table 1.** Number and percentage of river water bodies classified according to flow regime temporality for each national river type (NRT) set in the Catalan RBD (officially used according to the water framework directive (WFD)). Two different methods were compared in order to classify temporality: (i) natural flows prescribed regimes (NFPRs) calculated by using a rainfall-runoff model; and (ii) aquatic phases regimes (APRs) calculated by using TREHS software (from the TRivers LIFE + Project).

| WFD National River Types (NRTs) *(Number of Water Bodies)* | Temporality according to Natural Flows Prescribed Regimes (NFPRs) *(Number and % of Water Bodies)* | Temporality according to TREHS' Aquatic Phases Regimes (APRs) *(Number and % of Water Bodies)* |
|---|---|---|
| MHS-Siliceous wet mountain rivers (5) | Permanent (5–100%) | Perennial (5–100%) |
| MHC-Calcareous wet mountain rivers (12) | Permanent (12–100%) | Perennial (12–100%) |
| MMS-Siliceous Mediterranean mountain rivers (11) | Permanent (9–82%) Seasonal (2–8%) | Perennial (8–73%) Alternate-Fluent (3–27%) |
| MMC-Calcareous Mediterranean mountain rivers (34) | Permanent (28–82%) Seasonal (5–15%) Intermittent (1–3%) | Perennial (24–70%) Quasi-perennial (5–15%) Alternate-Fluent (5–15%) |
| MMEC-Mediterranean mountain rivers with high discharge (13) | Permanent (13–100%) | Perennial (13–100%) |
| RMCV-Lowlands dry Mediterranean climate rivers (113) | Permanent (59–52%) Seasonal (34–30%) Intermittent (18–16%) Ephemeral (2–2%) | Perennial (42–37%) Quasi-perennial (17–15%) Alternate-Fluent (40–35%) Fluent-stagnant (5–4%) Stagnant (2–2%) Alternate-stagnant (3–3%) Alternate (1–1%) Episodic (3–3%) |
| RMS-Siliceous dry Mediterranean Rivers (2) | Permanent (2–100%) | Perennial (1–50%) Alternate-Fluent (1–50%) |
| ZC-Karst feed rivers (16) | Permanent (13–81%) Seasonal (1–6%) Intermittent (2–13%) | Alternate-Fluent (3–19%) Fluent-stagnant (1–6%) |
| TL-Small coastal streams (32) | Permanent (2–6%) Seasonal (23–72%) Intermittent (4–13%) Ephemeral (3–9%) | Perennial (1–3%) Quasi-perennial (3–9%) Alternate-Fluent (15–47%) Alternate (1–3%) Stagnant (5–16%) Alternate-stagnant (1–3%) Occasional (2–6%) Episodic (4–13%) |
| EP-Large watercourses (10) | Permanent (10–100%) | Perennial (10–100%) |

In 2008, the Spanish Government, together with scientific advice, promoted an additional (complementary) river classification for TRs to better tackle water management challenges in relation to environmental flow regimes [43]. Four different river categories were proposed according to their natural flow intermittence assessed by using rainfall-runoff models: permanent (P), which are never dry; seasonal (S), in which flow ceases or dries (nearly zero flows) below 65 days per year, based on an annual average; intermittent (I), in which water flow accounts for 100 to 300 days per year; and ephemeral (E), where water flows less than around 100 days per year, based on an annual average (i.e., only after rain events). These four categories are named hereafter as "natural flows prescribed regimes" (NFPRs). In the Catalan RBD, the river flow for each water body was obtained through a rainfall-runoff model (Sacramento model) [44], which provided simulations of natural water discharge daily for each water body and for a period of 67 years (from 1940 to 2007). By using the outcomes of this model, water bodies were classified according to their natural temporality into the above-mentioned NFPRs. Thus, a total of 153 river water bodies were classified as permanent (P), out of 248 total water bodies in the Catalan RBD (62%); 25 as intermittent (I), which means 10% of the total water bodies; 65 as seasonal (S) (26%); and five as ephemeral (E) (2%). Permanent water bodies accounted for 62%, whereas TRs (considering seasonal, intermittent, and ephemeral) accounted for 38% of water bodies.

However, as stated above, rainfall-runoff models can overestimate low flows (close to zero flows), and cannot reliably detect zero flows or whether riverbeds are constituted

by disconnected pools or completely dry [8,10,37]. Therefore, an additional method to better classify TRs, which especially considers low and zero flow regimes, and changes in connected-disconnected pool patterns was required, especially because pools are ecologically unique and have high biodiversity values [45]. The TREHS software allows the classification of TRs taking into account three main axes (aquatic phases): flow permanence (Mf), disconnected pools permanence (Mp), and dry river permanence (Md); in a FPD (flow-pools-dry) plot (Figure 2), which aims at describing the main hydrological regime controls on biological communities [37]. TREHS proposes a classification of the TR regime that takes into account these three axes for a total of nine different "aquatic phases regimes" (APRs) (Figure 2): perennial (P), permanently flowing; quasi-perennial (Qp), usually flowing; fluent-stagnant (FS), occasionally drying with disconnected pools; alternate-fluent (AF), usually flowing and stagnant at infrequent intervals; stagnant (St), usually with disconnected pools but never dry; alternate-stagnant (AF), usually with disconnected pools that occasionally dry; alternate (A), rotates between disconnected pools and dry aquatic phases; occasional (Oc), usually dry; and episodic (Ep), with long dry periods (almost never flowing). The APR classification was designed to: (i) be fully applicable from obtainable information, (ii) take into account the statistics of the three aquatic phases, (iii) be represented in a single graph, (iv) be conflict-free from the most usual terminologies, and (v) be defined from hydrological features assumed to have biological implications. Data for this classification can be collected from several sources: gauging stations, model simulations, interviews or surveys, in situ observations, and aerial or ground-level photographs for a significant period [37].

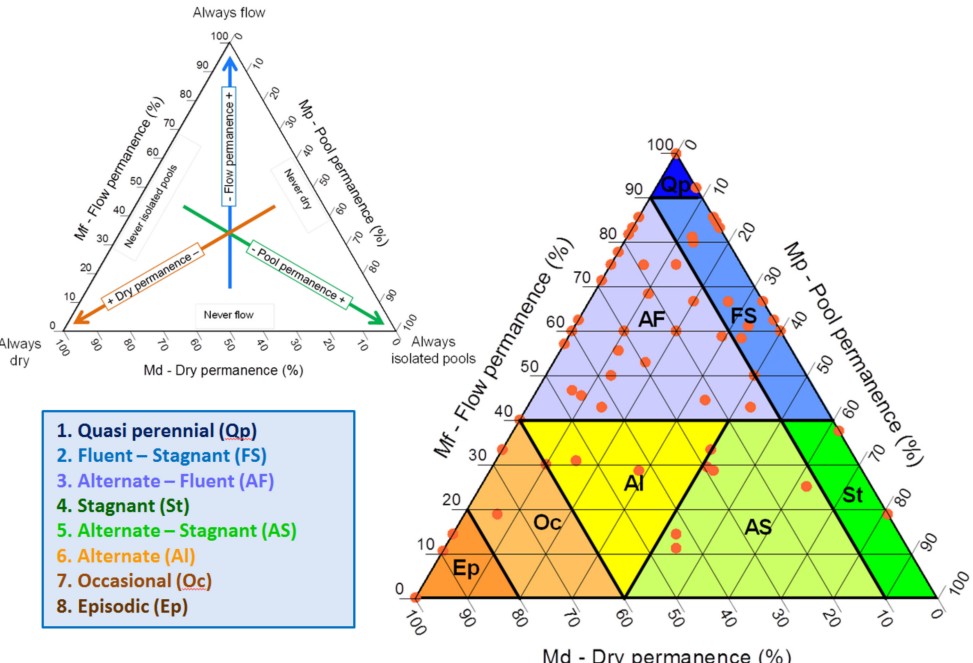

**Figure 2.** Distribution of the TREHS aquatic phases regimes in a flow-pools-dry (FPD) plot. Qp: quasi-perennial; AF: alternate-fluent; FS: fluent-stagnant; St: stagnant; AS: alternate-stagnant; Al: alternate; Oc: occasional; EP: episodic. Mf: flow permanence; Mp: pool permanence; Md: dry channel permanence. The orange dots represent river stations where the metrics were obtained from in situ or photographic observations (from [37]).

TREHS was tested in the Catalan RBD in order to properly classify the hydrological regime of TRs after visiting field sites, analyzing historical aerial photographs, and doing interviews with locals for historical and current river regimes, together with information provided from the Sacramento model for simulated natural flow regimes. The nine APRs were set for the Catalan RBD: 128 out of 248 water bodies were classified as perennial (51%), 25 as quasi-perennial (10%), six as fluent-stagnant (2%), 67 as alternate-fluent (27%),

seven as stagnant (3%), four as alternate-stagnant (2%), two as alternate (1%), two as occasional (1%), and seven as episodic (3%). Therefore, 51% of water bodies were classified as permanent, with 49% as temporary.

To compare outcomes from the different approaches in terms of TR classification in the Catalan RBD, the official NRTs were compared with NFPRs and with APRs (Table 1). The results showed that five NRTs were subdivided into additional NFPRs, showing the TRs (either seasonal, intermittent, or ephemeral) appeared in different NRTs: MMS—siliceous Mediterranean mountain rivers, MMC—calcareous Mediterranean mountain rivers, RMCV—lowlands dry Mediterranean climate rivers, ZC—karst feed rivers, and TL—small coastal streams. MMS and MMC river types grouped perennial and seasonal rivers, even some intermittent watercourses in MMC and ZC types, whereas RMCV and TL river types showed permanent, seasonal, intermittent, and even ephemeral watercourses in each river type. Almost 50% of RMCV rivers were non-perennial, and most of the TL river types behaved as temporary (mostly seasonal). Only MHS—siliceous wet mountain rivers, MHS—siliceous wet mountain rivers, MMEC—Mediterranean mountain rivers with high discharge, and EP—large watercourses, did not show different hydrological types within them, according to their temporality.

On the other hand, when TREHS was applied, six NRTs were subdivided into different APRs. Therefore, 27% of MMS water body types were classified as alternate-fluent, 15% of MMC as quasi-perennial, and 15% as alternate-fluent. RMCV, ZC, and TL river types showed a wide range of APRs within them. Over 60% of RMCV were classified as non-perennial, showing seven different APRs (quasi-perennial, alternate-fluent, fluent-stagnant, stagnant, alternate-stagnant, alternate, occasional, and episodic). Almost all TL water body types were classified as non-perennial (over 95%), whereas around 25% of ZC types showed alternate-fluent and fluent-stagnant APRs. RMS river type was divided into perennial and alternate-fluent APRs, whereas using NFPR type did not show differences according to natural river temporality.

Therefore, five out of 10 NRTs (50%) set in the Catalan RBD, by using the WFD Annex II methodology, showed differences when hydrological temporality was thoroughly analyzed by using the NFPR classification. On the other hand, when TREHS was applied in order to set APRs, six out of 10 NRTs showed differences within them, with a wider range of regimes (stagnant, fluent-stagnant, alternate, episodic, occasional, etc.). Relevant differences can be observed between the NFPR and APR classifications. When using NFPRs, minor differences were shown within MMS, MMC, RMCV, or ZC NRTs, whereas APRs gave much more hydrological differences, especially within MMC and RMCV, where stagnant and alternate-stagnant APRs can be clearly identified. Special attention has to be given to the TL NRT, where NFPR classified around 9% of them as ephemeral, whereas APR identified over 19% of them as occasional or episodic. Moreover, around 38% of water bodies were classified as temporary (considering seasonal, intermittent, and ephemeral) when NFPR classification was applied, whereas the percentage of temporary rivers surged to 51% when TREHS classification was used, and with a wider range of aquatic phases (APRs) in which connected or disconnected pool patterns were well identified. NFPR classification (based only on a rainfall-runoff model) seems not to be sensitive enough to low flows or even river dry up phases, and does not distinguish between connected and disconnected pools when flow is becoming scarce, or absent.

### 3.2. Effects of Non-Perennial River Classification When Applying Biological Quality Indices

As stated above, the main target for setting river typologies is to accurately assess biological quality according to reference conditions. Therefore, river types that better explain biological community patterns need to be used to properly classify ecological status. In order to analyze the different procedures for classifying TRs presented above, we compared the response of three biological quality indices used in the Catalan RBD: the IBMWP [46] and the IMMi-T [32] indices for macroinvertebrates, and the IPS index [47] for diatoms. Biological quality data were obtained from the monitoring program carried

out from 2013 to 2018 by the Catalan Water Agency. Samples were taken annually for all river water bodies (in the spring period), and values were assessed as EQR (ecological quality ratio) by dividing the biological quality values by the reference value according to each national river type (NRT). Thus, a total of six EQR values for 248 river water bodies were obtained for each biological quality index (data are available on the ACA webpage: http://aca.gencat.cat (accessed on 9 March 2021)), and the average of these six values for each site was calculated to get final biological quality values (IBMWP, IMMI-T, and IPS). These values were assessed along a stressor gradient considering chemical parameters (ammonia, nitrate, and phosphate) taken monthly from all river water bodies from 2013 to 2018. Additionally, human pressures for all river water bodies were taken into account according to the pressures and impact analysis (IMPRESS) carried out in the Catalan RBD [48]. Thus, 10 pressures were used to set the general degradation value for each river water body: density of weirs and dams; % of river straightened, and/or channelization; % of water withdrawal, flow regime alteration; % of human land uses in the floodplain area; % of urban wastewater discharge (purified and not purified); % of industrial wastewater discharge; % of agricultural use in the drainage area; % of mine activity in the drainage area. Each pressure was scored from 0 to 3 for each river water body (0: without pressure; 1: low pressure; 2: middle pressure; 3: high pressure) [48]. All three chemical parameters and the 10 pressures were combined to assess a single "degradation gradient" value. The average chemical parameters for each water body were standardized in order to range from 0 to 1. Then, the three chemical parameters were combined and weighted differently. Ammonia was weighted 3, phosphate 2, and nitrate 1. On the other hand, all 10 pressures were combined equally, and later standardized from 0 to 1. Finally, the "general degradation" gradient was obtained by combining the chemical gradient value, weighted 2, and the whole pressures gradient values, weighted 1, to get a range of values from 0 (low general degradation) to 1 (high general degradation) for all river water bodies. The general degradation value was calculated for all 248 river water bodies located in the Catalan RBD, even though these values were set in 233 out of 248 river water bodies, since 14 rivers did not have enough chemical data, due to the fact that they had remained completely dry for most sampling campaigns (mostly episodic/occasional or ephemeral watercourses).

Biological quality indices (IBMWP, IMMI-T, and IPS) were compared with the general degradation gradient by a lineal model, and the two TRs river classification procedures stated above (NFPR and APR) were plotted and visually compared (Figure 3). In order to be more comparable, and to make the chart more understandable, APRs were grouped into four "management temporary river categories" (MTRCs): perennial (includes perennial and quasi-perennial APRs); intermittent-fluent (includes fluent-stagnant and alternate-fluent APRs); intermittent-stagnant (includes stagnant, alternate-stagnant, and alternate APRs); and ephemeral (includes occasional and episodic APRs).

Largely, values of the three analyzed biological quality indices showed a good linear relationship with the general degradation gradient (Figure 3), which means they are WFD compliant [49]. Nevertheless, there was a certain dispersion of points around the regression line, due to some gaps in the response of the quality indices to the general degradation gradient. Preferably, all sites should be distributed as close as possible throughout the regression line, and gaps or values located far from the regression line need to be carefully analyzed in order to avoid misleading results due to an unsatisfactory adjustment of the index, or the inappropriate allocation to a particular river type, and/or the wrong reference values being used. Reviewing the relationship between each biological quality index and the general degradation gradient, we found differences when classifying TRs. When NFPR classification was applied (Figure 3), permanent and seasonal rivers were widely distributed around the regression line, and intermittent rivers were mainly located at the bottom of the regression line, even though quite a few are located above. Ephemeral rivers were not possible to plot, since general degradation values were not possible to assess due to lack of water to take biological and chemical data.

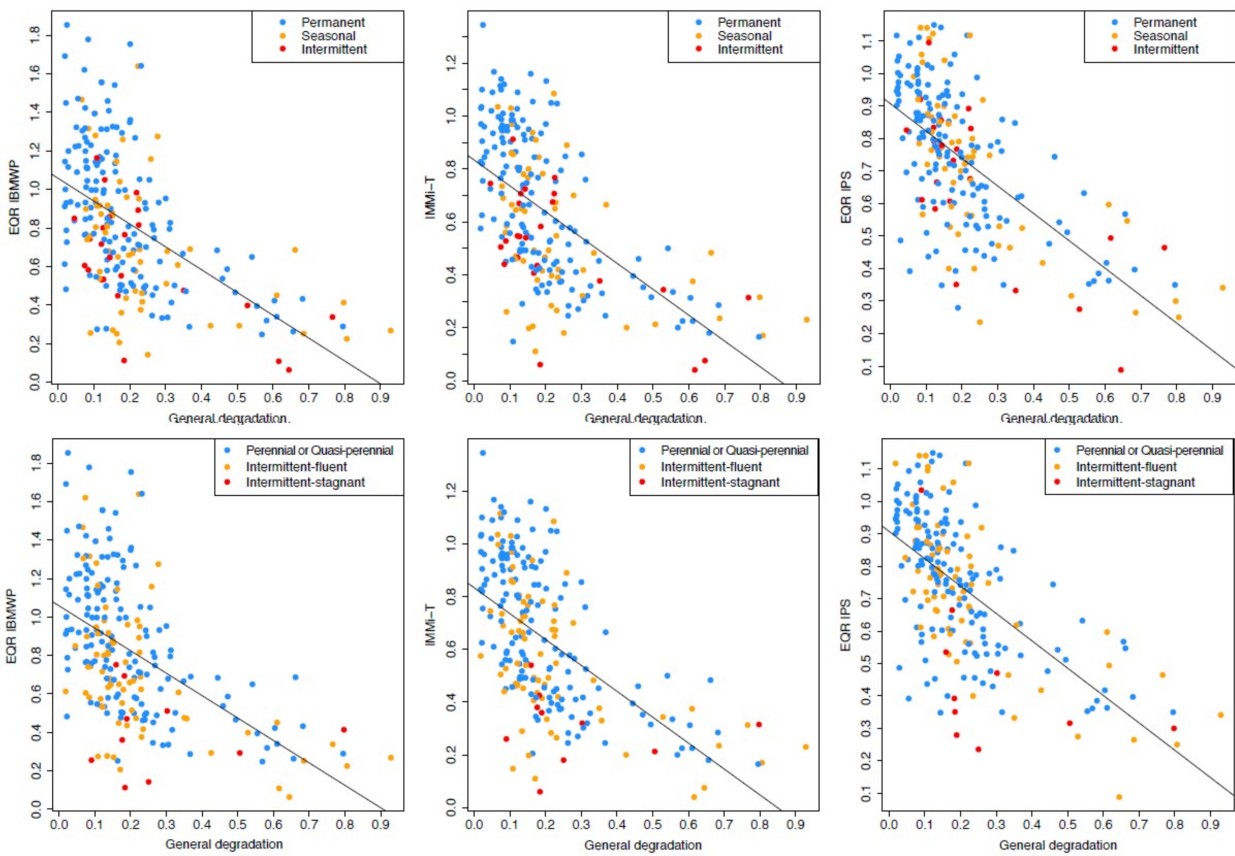

**Figure 3.** Biological quality indices (IBMWP, IMMI-T, and IPS) compared with a "general degradation" gradient. Two TR river classifications are showed: (i) "natural flows prescribed regimes" (NFPRs) established by using a rainfall-runoff model (above); and (ii) "management temporary river categories" (MTRCs) provided from the TREHS software (below).

Differences become much more relevant and clear when MTRCs were set by using TREHS (Figure 3). In this case, perennial or quasi-perennial, and most of the intermittent-fluent rivers, were spread around the regression line, but intermittent-stagnant were clearly located below and far from the regression line for the three biological quality indices analyzed (IBMWP, IMMI-T, and IPS). The fact that all intermittent-stagnant rivers were mostly located below the regression line denotes a clear deviation in the relationship between the index value and the general degradation gradient, which results in a misleading biological quality assessment. Therefore, these TRs are not adequately considered, and therefore might need to be separately classified, and have proper biological quality indices or/and reference conditions applied. Ephemeral rivers were not plotted due to a lack of biological and chemical samples. Therefore, whereas NFPR showed some mismatches between biological quality indices and the general degradation gradient when TRs are classified, MTRC classification by using APRs provided clearer differences, especially in identifying intermittent-stagnant rivers. Therefore, MTRCs provide a more useful TR classification for management purposes, in which intermittent-stagnant rivers can be sorted separately and then properly treated by using suitable biological indices and reference conditions.

Intermittent-stagnant rivers were easily discriminated by using MTRCs in some NRTs (Figure 4). Therefore, when MTRC classification was applied, intermittent-stagnant river types were identified in RMCV and TL NRTs, whereas the NFPR procedure was not able not distinguish them. MTRCs also provided a higher TR discrimination among NRTs, rather than NFPR. Thus, MMS, MMC, RMCV, ZC, and TL showed a much higher percentage of intermittent-fluent and ephemeral water bodies when applying MTRCs classification based on APRs, rather than applying NFPRs (Figure 4). Special attention needs to be given to the RMS river type, in which NFPRs did not distinguish any temporary water

bodies, whereas when MTRCs were applied around 50% of water bodies were classified as intermittent-fluent.

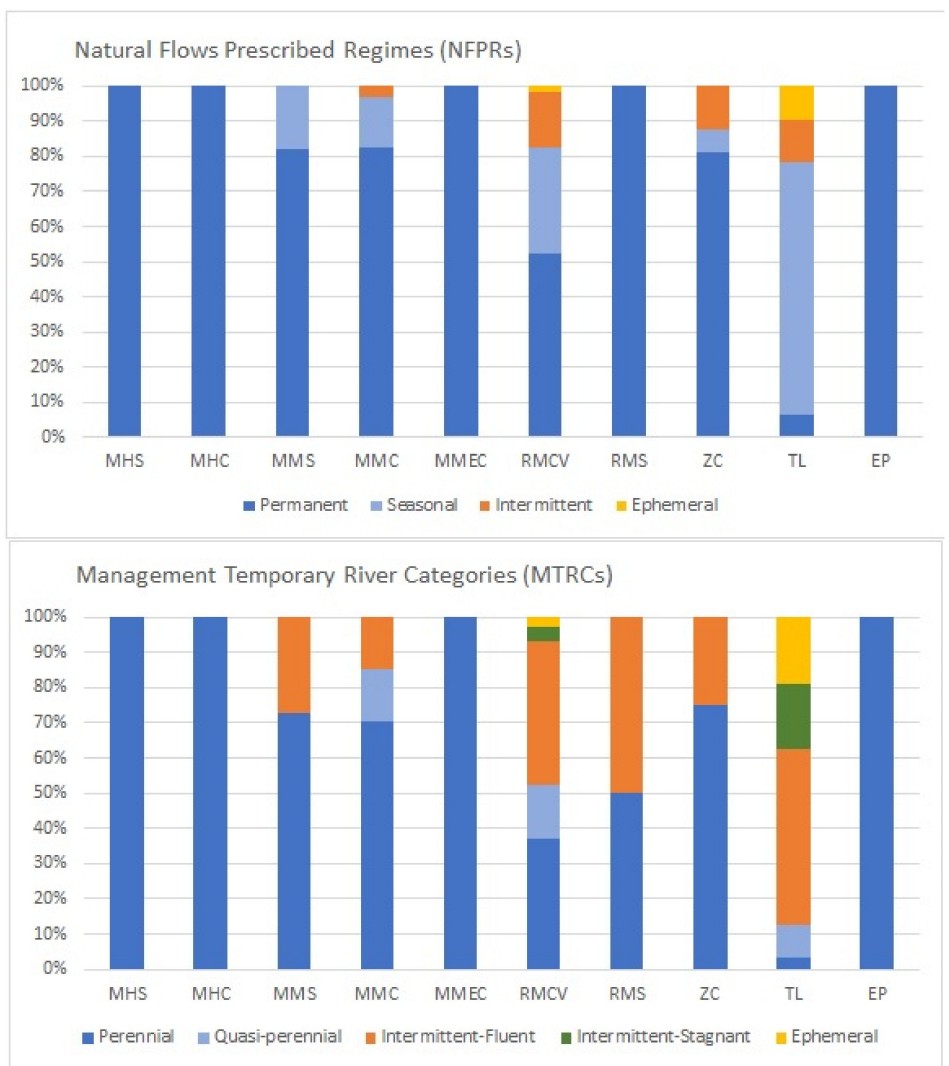

**Figure 4.** Percentage of river water bodies classified according to: (i) "natural flows prescribed regimes" (NFPRs) established by using a rainfall-runoff model (above); and (ii) "management temporary river categories" (MTRCs) provided from TREHS software (below), in each national river type (NRTs) set in the Catalan RBD.

It should be noted that in 14 out of 248 river water bodies in the Catalan RBD, biological quality was not able to be assessed due to lack of biological and chemical samples. No data were able to be gathered for a period of six years (2013–2028) due to the absence of water in the rivers, not even isolated pools. These water bodies were classified as ephemeral MTRC by TREHS, and alternative methods should be applied to classify their ecological status.

*3.3. Hydromorphological Quality Assessment in Ephemeral Watercourses*

A high percentage of river water bodies (23%) were classified as ephemeral in the Júcar RBD according to the NFPR classification. In 2017 and 2018, a monitoring program was conducted by the Júcar Water Authority in all water bodies which did not carry enough flow of water to allow biological quality assessment (ephemeral water bodies). A total of 70 water bodies were analyzed by using a hydro-geomorphological (HYMO) method. A hydro-geomorphological quality index adapted to TRs (IHG-E, [50]) was applied. The IHG-E index takes into account several elements to assess the river functional quality

(naturalness of the water flow, solid flow, and functionality in flood), the river bed structure (longitudinal naturalness and shape), and the riparian quality (longitudinal continuity, width of the corridor, and naturalness).

The results showed that 28 out of 70 water bodies (40%) were classified as having good or high hydro-geomorphological quality, whereas 42 water bodies (60%) were classified below good status, mostly moderate (Table 2). The worst valued hydro-geomorphological element measured was the quality of the river bed (Figure 5), since most of them remain dry for a long time, and are often altered by human activities due to lack of knowledge and scarce social awareness about these ephemeral ecosystems. This work allowed making suggestions to improve and draft the hydro-morphological procedure later adopted by the Spanish Government for TRs [51]. The hydro-morphological protocol, specially adapted for temporary and ephemeral rivers, is based on six functional elements: flow alteration and hydrology, connection with groundwater, river continuity, river morphology, structure and substrate of the river bed, and quality of the riparian zone. This protocol can be used in ephemeral MTRCs with a lack of biological and physicochemical data to assess ecological status properly.

**Table 2.** Hydro-geomorphological quality results by using the hydro-geomorphological quality (IHG-E) index [50] in ephemeral rivers in the Júcar RBD.

| HYMO (Hydro-Geomorphological) Quality | Num. of Water Bodies and Percentage | Water Body Length (km) and Percentage |
|---|---|---|
| High | 11 (18%) | 229.6 (16%) |
| Good | 17 (21%) | 278.37 (24%) |
| Moderate | 24 (39%) | 500.70 (34%) |
| Poor | 13 (18%) | 234.01 (19%) |
| Bad | 5 (4%) | 52.23 (7%) |

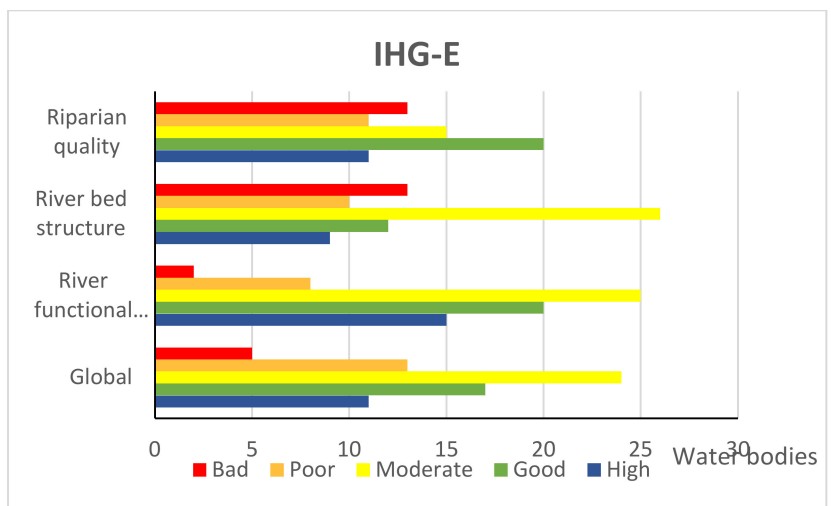

**Figure 5.** Number of river water bodies classified according to the Hydro-geomorphological quality index adapted to temporary rivers, IHG-E [50]. The global IHG-E results, and the three main quality elements: the river functional quality, river bed structure, and the riparian quality, are shown. The IHG-E index was applied to 70 ephemeral river water bodies in the Júcar RBD.

## 4. A Proposal of Temporary Rivers Classification for Ecological Status Assessment

According to the results stated above, we developed a common methodology to better classify TRs, in order to assess their ecological status for management purposes. Flow regime temporality and cease of flow are largely neglected in classifying river types, although, as widely demonstrated, they are relevant for determining the presence or survival of certain biological species and the reliability when applying biological quality indices [45,52,53]. Misleading results can be obtained if TRs are not considered and classified

correctly, paying special attention in intermittent-stagnant and ephemeral watercourses, and shifts among aquatic phases. Therefore, according to the experience gained from the Catalan and Júcar RBDs by the LIFE + TRivers Project, we propose a flowchart to better classify TRs and correctly assess their biological quality (Figure 6):

- First of all, natural flow regime conditions and changes over time need to be properly collected to classify TRs, and to assess the potential degree of hydrological alteration. Rainfall-runoff models can be used if they are accurately calibrated with recorded near-natural data from properly selected gauging stations, or corrected data by a complementary water allocation model. Nevertheless, as showed above, rain-fall-runoff models could not provide sensitive enough data, especially for low and zero flows, resulting in the underestimation of available aquatic habitats (i.e., connected or disconnected pools). Additional detailed information regarding flow regime and its changes over time is required and could be obtained through several sources, such as interviews of locals, field observations, and aerial photos, which can provide historical and useful information [29]. Thus, relevant information regarding the timing of flow cessation, and the presence of disconnected pools needs to be properly collected and entered into the TREHS software.

- Second, the flow regime has to be thoroughly checked to determine whether or not it has been altered by human activities. If a river is naturally perennial, and the flow intermittence is due to human activities, such as water abstraction resulting in severe hydrological impacts, water bodies should be classified as perennial, and biological quality indices defined for perennial river types should be applied accordingly. Conversely, some natural TRs can become permanent due to water discharge from human activities (e.g., urban wastewater treatment plants that discharge in a non-permanent river bed). In these cases, water bodies should be considered as TRs, and suitable ecological assessment procedure applied accordingly, or stated as heavily modified water bodies if near natural conditions cannot be restored affordably.

- Once river water bodies have been classified according to their natural flow regime, those identified as naturally TRs can be classified into different TR typologies using TREHS. The APR classification gives a good classification system for better managing non-perennial water bodies [37]. For the practical assessment of ecological status, some APRs can be further grouped, in order to apply common metrics and biological quality assessment procedures. In our case, we grouped the nine APRs into four MTRCs, which was easier for management purposes (Figure 6). Therefore, under perennial MTRC, the biological quality of the water body can be assessed according to current protocols developed for perennial rivers. On the other hand, when rivers are classified as intermittent-fluent MTRC, reference values need to be carefully reviewed, and new quality class boundaries established if necessary. When river regimes are classified as intermittent-fluent MTRC, and remain for a long time without flowing water, and with disconnected pools which almost never dry, new suitable quality indices and metrics need to be developed and applied. Finally, when water bodies are classified as ephemeral MTRC only hydromorphological indicators can be applied, or novel biological quality metrics can be developed, such as terrestrial biological community assemblages, terrestrial plants, or the aquatic invertebrate seedbanks [34,35].

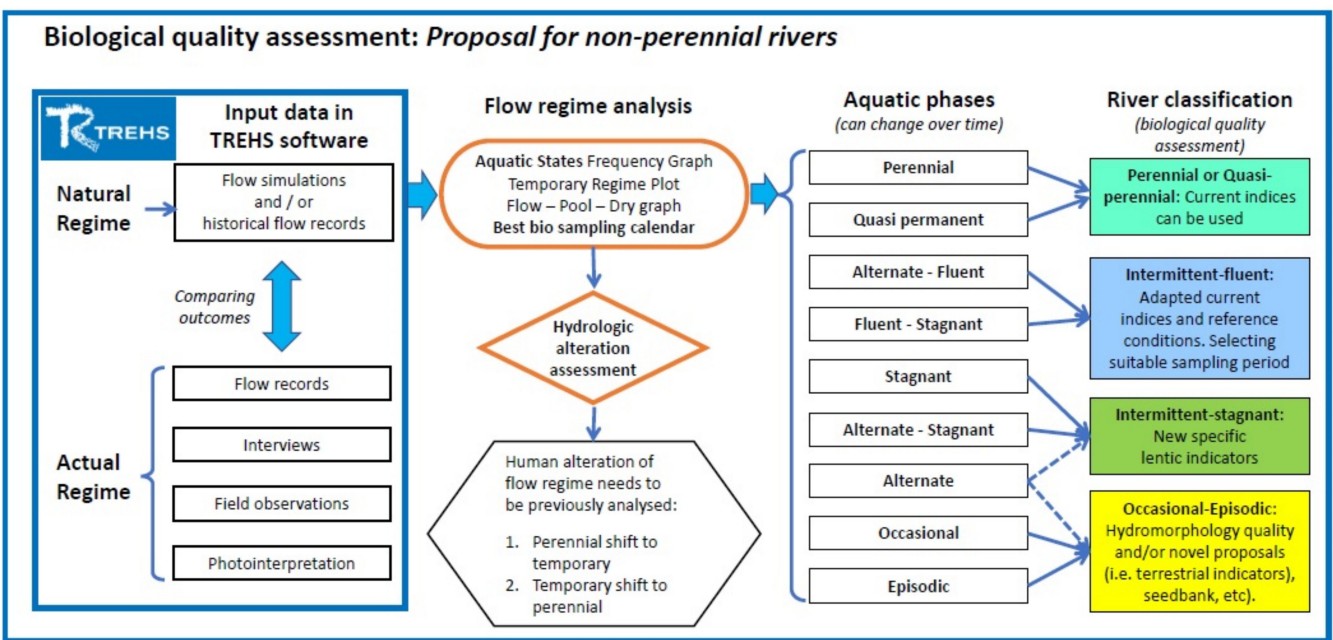

**Figure 6.** A proposed flowchart to better classify TRs, and correctly assess biological quality.

TREHS was designed to be used at the site (reach) scale, while flow intermittence is also a spatial issue because diverse regimes may coexist at the same time along a sufficiently long river section. Given its situation, the analysis of spatial patterns of flow intermittence must be handled properly, and biological status assessment must be calculated in a representative river reach from each water body. In all cases, it is paramount to properly collect information of previous aquatic states for each sampling site in order to choose the most suitable period to take representative samples. In this case, TREHS also provides a useful tool for this purpose [37].

The four MTRCs: perennial, intermittent-fluent, intermittent-stagnant, and ephemeral, defined in this paper can be slightly modified, and thresholds among APRs adapted according to water authorities needs and the biological indices they are using. Therefore, the Spanish Ministry just adopted a guide for better classifying TRs by using the method de-scribed in this paper [54] (Table 3) (Figure 7), in which thresholds among APRs were adapted into the four different NFPRs assessed by using rainfall-runoff model and officially used so far [43]. This novel procedure should be rolled out in coming RBMPs third cycle (2022–2027) by Spanish water authorities. Therefore, additional information will be available in the coming years to thoroughly analyze this classification.

**Table 3.** Criteria to set the four "management temporary river categories" (MTRCs): perennial or quasi-perennial, intermittent-fluent, intermittent-stagnant, and ephemeral, proposed in the Mediterranean Spanish basins [54] according to the three main axis (aquatic phases): flow permanence (Mf), disconnected pools permanence (Mp), and dry river permanence (Md) provided by TREHS software.

| Management Temporary River Categories (MTRCs) | % Flow Permanence (Mf) | % Pool Permanence (Mp) | % Dry Permanence (Md) |
|---|---|---|---|
| Perennial or Quasi-perennial | $99 < Mf \leq 100$ <br> $82 < Mf \leq 99$ | $0 \leq Mp < 1$ <br> $0 \leq Mp \leq 18$ | $0 \leq Md < 1$ <br> $0 \leq Md \leq 18$ |
| Intermittent-fluent | $27 < Mf \leq 82$ | $0 \leq Mp \leq 73$ | $0 \leq Md \leq 73$ |
| Intermittent-stagnant | $0 < Mf \leq 27$ | $40 \leq Mp \leq 100$ | $0 \leq Md \leq 60$ |
| Ephemeral | $0 < Mf \leq 27$ | $0 \leq Mp \leq 40$ | $33 \leq Md \leq 100$ |

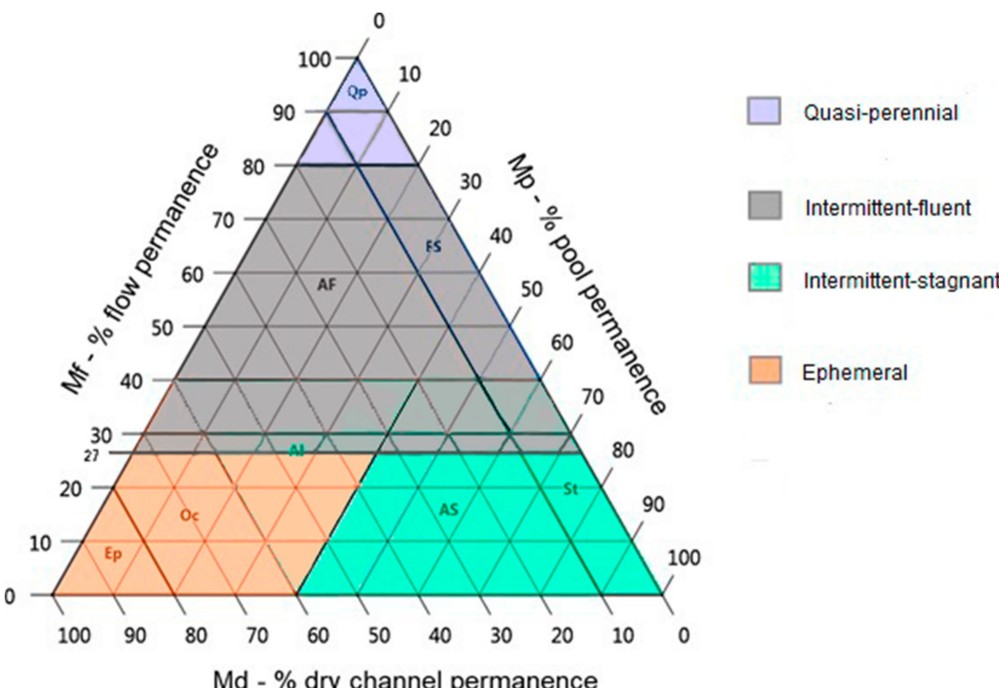

**Figure 7.** Four "management temporary river categories" (MTRCs): perennial and quasi-perennial, intermittent-fluent, intermittent-stagnant, and ephemeral, proposed for the Mediterranean Spanish basins [54], according to the three main axis (aquatic phases): flow permanence (Mf), disconnected pools permanence (Mp), and dry river permanence (Md) provided by TREHS software.

## 5. Discussion

The European WFD (2000/60/EC) [30] mandates that member states define river basin districts as the management framework, and designate water bodies, river types, and their reference conditions to properly assess their ecological status [25,40]. Measures to preserve or to achieve good ecological status have to be applied to water bodies according to their reference conditions [26,27]. Therefore, defining suitable river types becomes an essential first step in classifying the ecological status accurately, and to further apply effective measures. However, difficulties and shortcomings appear when water authorities try to identify and manage TRs since there is no clear classification pathways for management purposes.

Most European river basin authorities have defined river types according to Annex II of WFD in each RBMP, but hydrological parameters (i.e., flow regime temporality) do not appear with a relevant weight, or are completely absent in setting river types by using this method. This is likely one of the major reasons why TRs have not been clearly defined in most European RBMPs, and/or their ecological status has been under or overestimated due to improper use of biological quality indices and reference conditions (mainly set for perennial watercourses). Sánchez-Montoya et al. [55] demonstrated that the concordance between river types set by following WFD Annex II and macroinvertebrate community assemblages did not completely fit in some Mediterranean rivers. Similar results were obtained when predictive models of aquatic macroinvertebrate distribution were applied in Spanish Mediterranean rivers, in which biological patterns did not completely fit national river types, especially for temporary ones [56]. These, and other similar results [57] suggest that additional hydrological parameters should be taken into account when river types are defined, especially for TRs. One of the more relevant features of the water regime that affect biological communities are no-flow periods and rewetting [58,59], since the occurrence or disappearance of aquatic habitats depends on the water presence and flow. Thus, flow intermittence should be considered as a key variable when river types are set for management purposes, especially in RBDs where TRs are, or can become, abundant (e.g., Mediterranean RBDs).

As demonstrated above, TREHS software provides a useful and easy procedure to properly classify TRs, clearly differentiating intermittent-stagnant and ephemeral MTRCs (stagnant, alternate-stagnant, occasional, and episodic). Disconnected pools in TRs are transitional habitats of major ecological relevance, as they support aquatic ecosystems during no-flow periods [2,59]. For those water bodies classified as stagnant (St) or alternate-stagnant (AS) (intermittent-stagnant MTRC), new biological quality metrics need to account for the community changes since disconnection [45]. So, the identification of these TRs, and later the development of metrics should be prioritized in the agendas of water agencies where TRs are common. More attention is required on these rivers, not only because their frequency is expected to increase in the future, but also because they are key ecosystems for biodiversity conservation. They act as reservoirs of biodiversity, especially for organisms without strategies to cope with drying river beds (e.g., fish [60]) and significantly contribute to the recovery of communities during rewetting [61,62]. As rivers get fragmented and disconnected, pools appear, and the environmental conditions of these habitats and their biological communities change with time. Abiotic characteristics may become very harsh because of evaporation or the accumulation of organic matter (e.g., oxygen concentrations decrease, nutrients increase, and pH can become variable) [63,64]. Similarly, biotic changes by the arrival of colonizing species, mostly predators, impose extra challenges to biological communities. In some cases, these disconnected pools can persist for many months [65], and act as transitional habitats between lotic habitats. In other cases, disconnected pools dry up, and are transitional habitats between lotic and terrestrial habitats. The relative frequencies of the three aquatic phases (flow-pools-dry) provided by TREHS synthesize the hydrological controls on aquatic life, and can be used to define the regime of TRs [37]. Common metrics to assess the biological quality do not reliable account for these particular environmental conditions and their temporal community changes. The metrics should consider the community changes with time, and dynamic reference conditions should be established, depending on the time since disconnection. In this sense, trait-based functional metrics or metrics at a species level coming from metabarcoding approaches offer promising opportunities [45,66].

To address whether biological quality classification is comparable among European member states, the European Commission undertook an intercalibration exercise to reach a common quality standard when each member state uses different biological quality indices in different water body types [25,67]. The three analyzed indices used in this paper were intercalibrated successfully [68]. The European Commission set common water body types for the intercalibration exercise, named IC-types, in order to compare thresholds between quality classes calculated by using indices from member states, and applying a common biological quality index. A total of five different river types were set for Mediterranean rivers in a specific working group settled in the Mediterranean area (Mediterranean Geographic Intercalibration Group-MedGIG): RM1, RM2, RM4, and RM5. From all these IC Mediterranean river types, RM5 was defined as "temporary" among IC participants (France, Spain, Italy, Portugal, Cyprus, and Greece): high flow variability and even dry for some periods. However, little data were collected among Med-GIG member states from RM5 since most water bodies had been found dry or with scarce aquatic habitats when monitoring this IC type; neither biological nor physicochemical data. Most of the collected data for RM5 were obtained from seasonal quasi-perennial or fluent-stagnant watercourses; but extremely seasonal, intermittent with disconnected pools, or occasional water bodies were not properly represented or simply absent. Therefore, quality thresholds considering biological quality indices applied in highly temporary water courses were not possible to properly intercalibrate due to the lack of enough data. Thus, the RM5 river type used in IC would be equivalent to quasi-perennial or alternate-fluent APRs, according to the TREHS categories, or even fluent-stagnant, in which biological quality indices currently used by MS can be used properly by taking into account suitable monitoring periods. Nevertheless, new approaches are required to be applied when stagnant (St), alternate-stagnant (AS), occasional (Oc), or episodic (Ep) APRs are evaluated. Therefore,

an additional procedure for better classifying temporary rivers is needed, and the three axes (aquatic phases) provided by TRivers LIFE+: flow permanence (Mf), disconnected pools permanence (Mp), and dry river permanence (Md), summarized in the FPD (flow-pools-dry) plot (Figure 2) [37], can be used for this purpose.

This is an issue of utmost importance for water managers, especially considering the increase of TRs due to the coming effects of climate change. It is evidenced by a recent survey completed by representatives from 20 European countries, which identified different management challenges for assessing ecological status properly in TRs, together with best practice and priorities that should be undertaken for research on this topic [35]. The recent SMIRES (science and management of intermittent rivers and ephemeral streams) COST Action Project brought together researchers and water managers from over 25 EU and worldwide countries to address TR issues, and provided useful knowledge to better manage them [23]. Additionally, the European Commission recently rolled out an ad hoc task group under the ECOSTAT working group in order to cope with this challenge (currently underway), and some countries and water authorities are searching for novel methods, and trying proposals, to manage TRs [22,66,69–71]. This is the reason why advances in the field of adapting river classification and quality indices to TRs, such as the proposal provided in this article, should be taken into account in coming water management plans.

## 6. Conclusions

The biomonitoring methods implemented so far by water authorities are mostly developed for perennial rivers. This has resulted in many limitations regarding the management of TRs. Moreover, essential hydrological variables for TRs, such as flow variability or zero flows, are usually omitted when classifying river types. Therefore, TRs have been largely neglected for management purposes. In this way, we think that the MTRC classification of TRs (based on APRs) provided in this paper can give useful solutions, allowing distinguishing different habitats according to biological community patterns for management purposes, rather than other classifications only based on simulated flows by rainfall-runoff models, which do not take into account pool permanence and stagnant aquatic phases. Rainfall-runoff models are widely used by water authorities for water allocation in planning scenarios, but a wider perspective must be taken into account when assessing biological quality in TRs. Therefore, we propose using the TREHS flow-pools-dry plot [37], grouped into management temporary river categories (MTRCs) to better classify TRs, and then to use suitable protocols for each to assess ecological status. This proposal has some weaknesses, as TRs have a wide range of singularities, which change over time and along the river courses, so trying to classify them into a few management types can lead to underestimate singularities, and getting misleading results. In addition, the lack of data in small and temporary rivers can be identified as a threat to applying TREHS software properly and classifying TRs successfully. Nevertheless, the procedure we suggest in this paper provides a more suitable and realistic classification of temporary rivers, compared to most methods used so far, and water managers need to better manage them, especially in the coming climate change scenarios.

**Author Contributions:** N.B., N.C., M.S., F.G. and N.P. conceived the ideas of this manuscript, provided scientific advice and significantly contributed to the writing together with A.M. N.B. additionally contributed with some figures. C.S. (Carolina Solà), M.B., A.R., C.S. (Clara Sierra), P.F., P.L. and J.L. contributed preparing and analyzing data and getting results. T.E., A.F., I.S., S.J. and R.V. contributed providing information and data from Júcar RBD and HYMO analysis in temporary rivers, and criteria to set the temporary river categories for management purposes. All co-authors contributed to the drafts, and approved the submission. All authors have read and agreed to the published version of the manuscript.

**Funding:** This research is based on TRivers LIFE + Project outcomes (LIFE13 ENV/ES/000341). The Catalan and Júcar RBDs provided funds for monitoring and data treatment.

**Institutional Review Board Statement:** Not applicable.

**Informed Consent Statement:** Not applicable.

**Data Availability Statement:** Biological and chemical data from the Catalan Water Agency are available in the ACA webpage: http://aca.gencat.cat, accessed on 9 March 2021. Hydro-morphological results applied in ephemeral streams in the Júcar RBD can be found in www.chj.es, accessed on 9 March 2021. TREHS software and outcomes from TRivers LIFE+ Project are available in http://www.lifetrivers.eu, accessed on 9 March 2021. The MIRAGE FP7 Project (2009–2011) can be consulted in https://cordis.europa.eu/project/id/211732, accessed on 9 March 2021. COST Action Project and related papers can be found in https://www.smires.eu, accessed on 9 March 2021.

**Acknowledgments:** We appreciate the help provided by the TRivers LIFE + Project team setting up the protocols provided and collecting data. We also appreciate civil servers and river keepers from Catalan and Júcar RBDs who gathered field data in TRs. Special thanks to Gorka Muñoa for providing some figures and GIS work. We also appreciate discussions and advice from several researchers and water managers carried out in SMIRES Cost meetings.

**Conflicts of Interest:** The authors declare no conflicts of interest.

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
