# Peer review of "A Proposal to Classify and Assess Ecological Status in Mediterranean Temporary Rivers: Research Insights to Solve Management Needs"

_water, doi:10.3390/w13060767_

Round 1

Reviewer 1 Report

Antoni et al. proposes to classify and assess the ecological status in Mediterranean temporary rivers, stating that " biomonitoring methods implemented by water authorities are mostly developed for perennial rivers and do not apply to temporary river." proposing a better classification to asses the current ecological status.

The proposed chart for non-perrenial rivers is logic although not flawless.

Line 507, citing Sánchez-Montoya (citation [55] ) " concordance between river types set by following WFD Annex II and macroinvertebrate community assemblages did not completely fit in some Mediterranean rivers"  however, zoobentos remains a classic indicator that can not be disregarded and at least one more citation as [55] is required in the defense of your paper.

Author Response

We really appreciate your comments and suggestions to enhance the manuscript.

Regarding the first comment (proposed chart logic but not flawless). We understand that the proposed flowchart to classify Temporary Rivers is not completely flawless, but enhance the current proposals used so far by Water Authorities, mainly based on perennial rivers. Our proposal arises from the MIRAGE FP7 project and the LIFE+ TRivers, so we think that it relays on a scientific knowledge translated to management needs. Other countries with similar percentage of temporary and ephemeral rivers (Cyprus, Malta, Portugal, Greece, etc.) are testing or trying to apply similar proposals as well in order to better classify this kind of rivers. We changed a bit the title of the manuscript.

Regarding the second comment (Line 507): We included additional citations to reinforce this sentence as the reviewer suggested. Thus, we added: “Similar results were obtained when predictive models of aquatic macroinvertebrate distribution were applied in the Spanish Mediterranean rivers, in which biological patterns did not completely fit into national river types, especially for temporary ones [56]. These, and other similar results [e.g. 57],…” (See Lines 509-513).

Thanks for your comments.

Reviewer 2 Report

Paper is dealing with quite novel methodology for categorisation of the temporary rivers in Mediterranean area. Materials are analyzed in detail. The guiding thought is clearly expressed throughout the paper. Despite this, I am suggesting a major revision.

1. Authors should explain which countries they mean by ''Mediterian'' countries? What do they mean by this adjective? Map should be enclosed.

2. Authors should give insight how different countries will (and if they will) accept such concepts? Everything is explained very well, the content of table 1 is presented in detail, but I must admit that I am very sceptical about it.

3. Authors should comment on the relationship with the different approaches for the biological minimum. There are many approaches in the EU countries, but if such approaches cannot be defined as a rule, but as a suggestion, I have the same concerns about presented methodology. I am referring to the point 2 of my revision.

4. I know from my scientific, professional and teaching experience that such a complex problem is very hard to elaborate very shortly in Conclusion. A strong SWOT analysis should be provided in the section before Conclusion, so that the Conclusion should be truncated. 

Author Response

We really appreciate your comments and suggestions to enhance the manuscript.

Regarding the first comment (Authors should explain which countries they mean by ''Mediterranean'' countries - map should be enclosed), we believe that incorporating a map in a new figure can give a disproportionate view, since we only use the concept of “Mediterranean” as a climatic reference in the manuscript. Thus, notice that we are referring to “Mediterranean climate region”, rather than "Mediterranean countries" which is related to the climate zone located around the Mediterranean Sea. We define the Mediterranean climate area in the “Study area” section as: “The TRivers project was conducted in two Mediterranean River Basin Districts (RBD): the Catalan RBD and the Júcar RBD located in NE and E Spain, respectively. Both RBDs have Mediterranean climate, with a precipitation ranging from 400 to 750 mm/year, highly irregular among years (± 200). Low water flows and dry watercourses are usual, particularly in summer, together with sudden floods after heavy rains”. In order to solve the Reviewer comment, we have included “close to the Mediterranean Sea” in this sentence (see Line 125 in the manuscript) in order to clarify.

Regarding the second comment: As we already say in the title of the manuscript, we provide only a proposal which aims to ease the management of temporary rivers by classifying them better. Water managers usually manage perennial rivers (with continuous water flow) but river intermittence provides new challenges that need to cope with properly. Most co-authors of this manuscript were previously involved in MIRAGE FP7 project and the LIFE+ TRivers base on temporary rivers, which provide us good scientific knowledge to better manage temporary rivers. Therefore, in this paper, we propose a novel procedure obtained from these research outcomes. Other countries located around the Mediterranean area have quite similar issues when managing non-perennial rivers, especially for ephemeral ones (e.g., Cyprus, Malta, Portugal, Greece, etc.). So, we propose a new procedure, a novel pathway to better classify temporary rivers and apply suitable biological quality methods to assess its ecological status. It’s only information that we offer to water managers and scientist that obviously they can apply or not, or even enhance it.

Regarding the third comment: We agree with the sentence of the Reviewer regarding “there are many approaches in the EU countries” and we likely "need a rule rather than a suggestion" to be applied successfully. That is why we are closely working together with the European Commission, and the ECOSTAT Group which is in charge of reviewing rules and provide “official” guides to better classify and assess ecological status in the European water bodies. Some co-authors of this manuscript we are involved in the ECOSTAT group regarding temporary rivers, and we think that this paper can help to reach a common agreement among European Member States on this issue. We explain this in the manuscript (see Lines 604 and 605): “the European Commission rolled out recently an ad-hoc task group under the ECOSTAT working group in order to cope with this challenge (currently underway), and some countries and water authorities are searching for novel methods and trying proposals to man-age TRs - e.g., [22, 66, 69, 70, 71]. This is the reason why advances in the field of adapting river classification and quality indices to TRs, such as the proposal provided in this article, should be taken into account in coming water management plans”. Obviously, this manuscript provides a first step (a proposal) than need to be tested and later validated.

Regarding the fourth comment: According to the Reviewer comments, we added additional text and a SWOT analysis in the Discussion section:

  • STRENGTHS: This method provides a better and suitable classification of temporary rivers to assess ecological status properly.
  • WEAKNESSES: Temporary rivers have a wide range of situations and singularities, which change over time and alongside the river courses, so trying to classify them into specific and quite a few management types can lead to underestimate singularities and get misleading results.
  • OPPORTUNITIES: The method provided in this manuscript can be applied in other Mediterranean basins and even to non-Mediterranean areas with similar issues (temporary rivers are not only located in the Mediterranean area).
  • THREATS: Lack of data in small rivers to apply the methodology we propose.

We included this SWOT analysis in the Conclusions section (see Lines 609-617): “This proposal has some weaknesses, as TRs have a wide range of singularities, which change over time and alongside the river courses, so trying to classify them into a few management types can lead to underestimate singularities and get misleading results. Also, the lack of data in small and temporary rivers can be identified as a threat to apply TREHS software properly and classify TRs successfully. Nevertheless, the procedure we suggest in this paper provides a more suitable and realistic classification of temporary rivers, rather than most of methods used so far, which water managers need to better manage them, especially in coming climate change scenarios. Our proposal can be applied in other Mediterranean or non-Mediterranean areas with similar issues”.

Many thanks for your comments.

Round 2

Reviewer 2 Report

I am very satisfied how the manuscript was improved. Also, author's answers are fine. I am suggesting further procedure for publishing the paper.